# Energy and Resource Efficiency in Apatite-Nepheline Ore Waste Processing Using the Digital Twin Approach

**Maksim Dli [1], Andrei Puchkov [1,*], Valery Meshalkin [2], Ildar Abdeev [3], Rail Saitov [3] and Rinat Abdeev [3]**

1 Department of Information Technologies in Economics and Management, National Research University (Moscow Power Engineering Institute, Smolensk Branch), 214013 Smolensk, Russia; midli@mail.ru

2 Department of Logistics and Economic Informatics, Mendeleev University of Chemical Technology, 125993 Moscow, Russia; clogist@muctr.ru

3 Department of Technological Machines and Equipment, Bashkir State University, 450076 Ufa, Russia; air@bgutmo.ru (I.A.); ri@bgutmo.ru (R.S.); arg@bgutmo.ru (R.A.)

* Correspondence: putchkov63@mail.ru

**Abstract:** The paper presents a structure of the digital environment as an integral part of the "digital twin" technology, and stipulates the research to be carried out towards an energy and recourse efficiency technology assessment of phosphorus production from apatite-nepheline ore waste. The problem with their processing is acute in the regions of the Russian Arctic shelf, where a large number of mining and processing plants are concentrated; therefore, the study and creation of energy-efficient systems for ore waste disposal is an urgent scientific problem. The subject of the study is the infoware for monitoring phosphorus production. The applied study methods are based on systems theory and system analysis, technical cybernetics, machine learning technologies as well as numerical experiments. The usage of "digital twin" elements to increase the energy and resource efficiency of phosphorus production is determined by the desire to minimize the costs of production modernization by introducing advanced algorithms and computer architectures. The algorithmic part of the proposed tools for energy and resource efficiency optimization is based on the deep neural network apparatus and a previously developed mathematical description of the thermophysical, thermodynamic, chemical, and hydrodynamic processes occurring in the phosphorus production system. The ensemble application of deep neural networks allows for multichannel control over the phosphorus technology process and the implementation of continuous additional training for the networks during the technological system operation, creating a high-precision digital copy, which is used to determine control actions and optimize energy and resource consumption. Algorithmic and software elements are developed for the digital environment, and the results of simulation experiments are presented. The main contribution of the conducted research consists of the proposed structure for technological information processing to optimize the phosphorus production system according to the criteria of energy and resource efficiency, as well as the developed software that implements the optimization parameters of this system.

**Keywords:** digital twin; computational intelligence for modeling and control; apatite-nepheline ore waste processing; energy and resource efficiency

## 1. Introduction

Designing technical devices requires solving optimization problems, one of which is the problem of minimizing energy and resource consumption. Its solution allows for accurately predicting economic effects, but in each case, it takes into account the specificities of the subject area for which the device is created. Successful practices to increase energy and resource efficiency are based on the use of combined (hybrid) solutions using several energy sources [1], improving the architecture of devices [2]; for large-scale industries, the trilemma of energy, economic and environmental efficiency always has to be solved [3]. An example is the processing of apatite-nepheline ore waste, which accumulates in huge quantities in the tailing dumps of mining and processing plants and poses a significant environmental hazard to the adjacent territories.

One of the directions for processing apatite-nepheline ore waste is the production of yellow phosphorus from them [4]. A chemical and energy technological system (CETS) implementing the process of phosphorus production consists of three sequentially arranged units: a granulator (GR), a multichamber indurating machine of a conveyer type (MIMCT), and an ore thermal furnace (OTF). Figure 1 shows the scheme for the CETS. The granulator forms raw pellets from the apatite-nepheline ore waste; the indurating machine removes the excessive moisture due to the hot gas passing through the pellets' multilayer mass, and in the OTF the pellets are melted, releasing gaseous phosphorus.

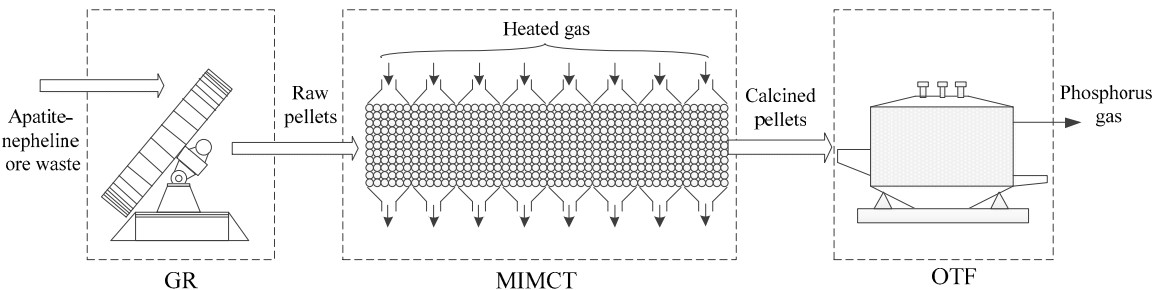

**Figure 1.** Scheme for chemical and energy technological system (CETS) of phosphorus production from apatite-nepheline ore waste.

The large volumes of raw materials, heat, and electrical energy consumed by CETS mean that even a relatively small decrease in them leads to significant economic effects in absolute terms [5]. This makes the problem of studying and developing tools to increase the energy and resource efficiency of CETS for phosphorus production from apatite-nepheline ore waste urgent; one of the tools proposed in this work is based on the "digital twin" technology (Digital Twin, DT).

DT technology is a characteristic trend of the 4th industrial revolution (Industry 4.0) and means the constructionof an interactive digital model with a high degree of adequacy to real processes, which significantly speeds up the analysis of the effectiveness of the decisions made and the assessment of their consequences [6]. Some IT companies, such as SAP, IBM, and Oracle, supply market solutions based on DT technologies; the same concept is used by the industrial corporations Airbus, General Electric, Siemens, Boeing, and others. The concept is based on the assumption that for each physical system it is possible to create a virtual "mirror" image containing all information about the physical system. The DT structure contains various levels: the local data level, the IoT Gateway level, cloud databases for emulation, and simulations of real objects. It should be noted that the structure can be implemented in both modern and outdated manufacturing entities with minimal changes to the existing aggregates [7]. DT demonstrates the great potential for implementing a cyberphysical manufacturing system in the epoch of Industry 4.0 [8–10].

DT concept distinguishes between the following types:

- Digital Twin Prototype (DTP) contains a high-precision model of a real object, but at the same time does not include measurement results and reports coming from it.

- Digital Twin Instance (DTI) describes a real object and includes information about the model settings, control parameters, sensor readings, and history process. The field of DTI application is concerned with the forecasting of a real object state. DTI, unlike DTP, changes during the operation of a real process or a system.
- Digital Twin Aggregate (DTA) is a system used with DTI; they can query information and exchange data with each other.

System modeling of the entire technological system as a whole, and not of its individual parts, leads to the DT hybrid concept [11].

Additionally, the concept of the "digital environment" (Digital Twin Environment, DTE) is defined as a set of conditions and means for multidisciplinary, multiphysics, and multiscale studies directed at DT development.

The use of DTE in the design of automated process control systems (APCSs) reduces the cost of field experiments, leading to qualitatively new approaches to the formation and processing of technological information. These approaches are expressed, first of all, in the expansion of the range of information channels based on the use of the Industrial Internet of Things (IIoT), an increase in the speed of its updating and a significant increase in volume, which leads to the formation of technological Big Data and the expediency of using Big Data analytics [12]. Big Data tools have replaced CALS (Continuous Acquisition and Life Cycle Support) and PLM (Product Lifecycle Management) technologies for creating virtual production. The qualitative changes in modern computing technology contributed to this through a significant increase in the speed and depth of information processing of inexpensive computer systems via new architectures of processors, parallel computing, and intelligent methods of data processing. Classical statistical methods do not provide the full information that can be extracted from Big Data and applied in practice. Technology data can offer much more in terms of modeling and simulation. In particular, statistical methods do not cover the entire spectrum of No-factors (no clarity, no completeness, no correctness, and others) that can be presented in Big Data, the use of which gives the opportunity to get additional knowledge about technological process. Big Data processing within the DTE technology is based on the use of a variety of computational intelligence methods for modeling and control in technological processes that allow for working with No-factors.

Reengineering of APCS infoware, directed to use DTE, allows for obtaining significant benefits at minimal cost due to the optimization of operating conditions, and the technological process control, since the cost of hardware and software updating for APCS information support is much lower than the cost of technological units and systems.

The main contribution of this work lies in the proposed structure for processing data on the state of CETS and DTE software, which allows for optimizing the operating modes of CETS according to the criteria of heat and minimum electric energy consumption. The paper is organized as follows: Section 2 develops the theoretical DTI model and the resulting optimization problem, Section 3 contains the structure for the DTE software and shows the obtained numerical results, and Section 4 presents the study conclusions.

## 2. Materials and Methods

The technological processes involved in CETS differ in terms of their multidisciplinary, multiphysics, and multiscale nature, which makes obtaining a unified model of the entire CETS complicated and time-consuming. However, mathematical models are always based on certain assumptions, simplifications, and restrictions arising from the applied formal description of the subject area, which leads to errors in the results. In addition, unaccounted and random factors always have an impact on the results. In these conditions, it is reasonable to apply analysis data methods that do not require additional costs for improving the mathematical apparatus of the processes models, but allow for using the developed sensors network of various parameters set onthe stationary and dynamic units, and the increased capabilities of computing and telecommunication systems included in APCS.

Let us consider the energy and resource consumption optimization of CETS (hereinafter, energy consumption will mean specific energy consumption, measured in MJ/t) to produce phosphorus from apatite-nepheline ore waste. The optimization criterion can be written in a formalized form:

$$E_{\sum} = C_{EL}\, Q_{EL\_\sum} + C_H\, Q_{H\_\sum}, \tag{1}$$

where $E_{\sum}$ = total energy consumption; $C_{EL}$ = unit cost for electrical energy; $C_H$ = unit cost for heat energy; $Q_{EL\_\sum}$ = total consumption of electrical energy; and $Q_{H\_\sum}$ = total consumption of heat energy.

Energy and resource efficiency for the processing technology of apatite-nepheline ore waste means the CETS state at which Equation (1) is minimized.

The total consumption of electrical energy is the sum of the consumption of CETS individual units:

$$Q_{EL\_\sum} = Q_{EL}{}^{GR} + Q_{EL}{}^{MIMCT} + Q_{EL}{}^{OTF},$$

where $Q_{EL}{}^{GR}$, $Q_{EL}{}^{MIMCT}$, $Q_{EL}{}^{OTF}$ are the electrical energy consumption in the granulator, MIMCT, and OTF, respectively.

Heat energy consumption is concentrated in the MIMCT, containing a set of $n$ vacuum chambers to remove moisture from pellets and their roasting, so the total heat energy consumption is a sum:

$$Q_{H\_\sum} = Q_{H\,1}{}^{MIMCT} + Q_{H\,1}{}^{MIMCT} + \ldots + Q_{Hn}{}^{MIMCT},$$

where $Q_{Hi}{}^{MIMCT}$ = heat energy consumption by the $i$th vacuum chamber, $i = 1, 2, \ldots, n$.

Let us define two possible types of control: $U^c$ = the control, including solutions for the development and implementation of the algorithms of processes with optimal control taking place in CETS; $U^d$ = the control connected with the specification of the design parameters for the elements of this system, providing optimal energy and resource consumption. In other words, $U^c$ corresponds to the CETS parametric optimization, and $U^d$ corresponds to the structural one. Then, Equation (1) can be expressed by the following function:

$$E_{\sum} = F_1(U_{GR}{}^c, U_{MIMCT}{}^c, U_{OTF}{}^c, V_1, V_2, V_3) + F_2(U_G{}^d, U_{MIMCT}{}^d, U_{OTF}{}^d), \tag{2}$$

where $F_1()$ = functional reflecting the influence of optimal control and technological processes parameters $V_1$, $V_2$, $V_3$ = vectors for the CETS technological units of the GR, MIMCT, and OTF, respectively; $F_2()$ = functional reflecting design optimization solutions based on the exergic analysis of the CETS [13,14]:

$$F_2(U_{GR}{}^d, U_{MIMCT}{}^d, U_{OTF}{}^d) = \min \sum_{j=1}^{J} \left( (E_1{}^j - E_2{}^j)\, C_j + C_j{}^d \right),$$

where $J$ = the number of the potential sources for the heat recuperation, $E_1{}^j$ = all the energy that can be reused from the $j$th source, $E_2{}^j$ = energy of the $j$th source that is used at the current time; $C_j$ = the unit cost of the energy, and $C_j{}^d$ = the cost of additional solutions on construction optimization of the $j$th source.

Due to constraint fulfillment in CETS, presented in the form of inequalities, when optimizing efficiency criterion (1), high-quality implementation of chemical power engineering processes is achieved by reducing the return part, which ensures resource savings [15,16].

The optimization problem is in the minimization (2) under the condition to provide the required product quality $\gamma_P$, the degree of phosphorus purity at the output of OTF: $\gamma_{P\_giv} \leq \gamma_P$, where $\gamma_{P\_giv}$ is the given degree of the output product.

The vectors for the parameters of the CETS technological units have the following composition:

- The vector for the granulator parameters: $V_1 = (G_w, G_c, \alpha, U_T, V^1_{\text{TLG}})^{\text{T}}$, where $G_w$ = the water mass flow; $G_c$ = the raw material mass flow at the output; $\alpha$ = the plate's angle of inclination; $U_T$ = the electric motor voltage supply for the plate's drive; and $V^1_{\text{TLG}}$ = the physicochemical, granulometric, lithological, and thermophysical characteristics of the raw materials at the entrance to the granulator.

- The vector for the MIMCT parameters: $V_2 = (G_k, u_0, d, T^0_{g1}, \ldots, T^0_{gn}, V^2_{\text{TLG}}, T_{gi}, \ldots, T_{gn}, W_{gi}, \ldots, W_{gn})^{\text{T}}$, where $G_k$ = the pellets' mass flow rate at the MOMCT entrance; $u_0$ = the mean of the pellets' moisture content along the height of the multilayer bed at the drying zone exit; $d$ = the mean diameter of a pellet; $T^0_{gi}$ = the heat carrier gas temperature at the entrance of the $i$th vacuum chamber, $i = 1, \ldots, n$, where $n$ = the number of chambers, $V^2_{\text{TLG}}$ = the physicochemical, granulometric, lithological, and thermophysical characteristics of the raw materials at the entrance to the MIMCT; $T_{gi}$ = the air temperature at the exit of the $i$th vacuum chamber, and $W_{gi}$ = the air consumption in the $i$th vacuum chamber;

- The vector for the OTF parameters: $V_3 = (h_o, l_o, G_{ko}, \sigma_{\text{K}}, \eta_{\text{K}}, G_{\text{p}}, \gamma_{\text{p}}, V^3_{\text{TLG}})^T$, where $h_o$ = the height of the pellet layer, $l_o$ = the width of the pellet layer; $G_{ko}$ = the pellets' mass flow rate; $\sigma_{\text{K}}$ = the ultimate strength of the pellets; $\eta_{\text{K}}$ = the degree of reaction for the dissociation reaction of carbonates; $G_{\text{p}}$ = the phosphorus mass flow rate; $\gamma_{\text{p}}$ = the degree of phosphorus purity; $V^3_{\text{TLG}}$ = the physicochemical, granulometric, lithological, and thermophysical characteristics of the raw materials at the entrance to the OTF.

Vectors $V^1_{\text{TLG}}$, $V^2_{\text{TLG}}$, and $V^3_{\text{TLG}}$ reflect the physicochemical, granulometric, lithological, and thermophysical characteristics of raw materials, and can contain different numbers of components depending on the model requirements for a particular unit.

Resource consumption optimization is an important condition for ensuring the economic efficiency of various technological systems [17–21]. The optimization of energy and resource consumption in individual CETS units for the production of phosphorus is the subject of works by a number of authors, e.g., [5,15,16]. However, the solution for the criterion (2) minimization based on the application of classical mathematical approaches to the entire CETS, but not its individual parts, leads to the problem of inconsistency in the description and limitations of the models of individual units. In addition, the presence of a large number of parameters in Equation (2) makes it difficult to obtain a unified analytical description of the optimization procedure for the entire CETS. Under these conditions, a fundamentally new approach was proposed based on powerful computational and algorithmic support from modern digital technologies, DT technology in particular. Taking into account the significant cost of DTP, DTI, and DTA development, it was decided, first, to create a structure and software for DTE based on the apparatus of deep recurrent (Recurrent Neural Network, RNN) and convolutional neural networks (CNNs). The choice of these architectures is due to their high competitiveness in the analysis of multivariate time series, where longer forecast horizons are required, while autoregressive models are a good choice for relatively small datasets [22–24]. Such networks, realized in a productive computing system, are now the main direction for the practical application of the artificial intelligence methods in the monitoring and diagnostics of various systems. RNN and CNN solve a wide range of problems related to classification or regression analysis in the power engineering and chemical industries [25–29]. In the application under consideration, DNN will allow for automatizing the process of receiving and interpreting incoming multichannel technological information from CETS, large volumes of which, received due to the monitoring of the processes over a long time, reduce the probability of networks relearning.

Ensemble application of RNN and CNN makes it possible to forecast energy and resource consumption for time $T_{delay}$ due to the historical analysis of interrelated data on the technological process using RNN, and recognize the current and future state of CETS using CNN based on images generated from RNN output data. Moreover, when using several CNN, an additional possibility is created to process data coming from video surveillance systems for a technological process and to

conduct video analytics on that basis. The proposed DTA structure to process technological information based on CNN and LSTM is shown in Figure 2.

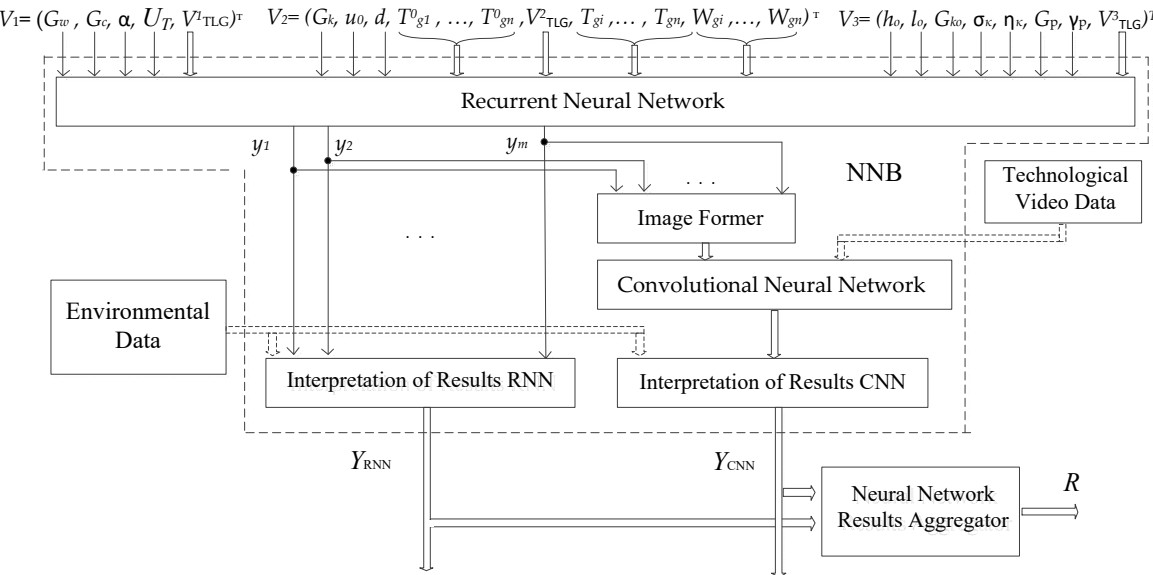

**Figure 2.** Structure of the processing of technological information in Digital Twin Environment (DTE).

In Figure 2, one cascade of a Neural Network Block (NNB) is built on the basis of Long Short-Term Memory (LSTM) recurrent networks having a high representative power in the processing of data sequences and their forecast [30]. The LSTM input receives a multichannel data flow consisting of vectors $V_1$, $V_2$, and $V_3$ taken at intervals $\Delta t$ during the time $T$ look from the current moment $t$. The multichannel LSTM input data format provides an accounting for the mutual influence of the technological parameters included in these vectors. $T_{\text{look}} = k_{\text{looc}}\Delta t$ determines the depth of the historical analysis performed by LSTM at the present, where $k_{\text{looc}}$ is the number of discrete historical analyses.

At the LSTM output a sequence $y_i = y(t - i\Delta t)$, $i = 0, 1, \ldots, m$ is created for each time point $t$. LSTM network is learnt on datasets whose structure is in the form of [*input* = {$V_1(t)$, $V_2(t)$, $V_3(t)$}; *output* = $E_\Sigma(t + T\_delay)$] for time point $t$. When $T\_delay = 0$, the network has learned to determine the current value of the criterion (2).

A set from $k_{CNN}$ of rows $y_i$ for discrete time moments from the interval $[t - T_{\text{CNN}}, t]$, where $T_{\text{CNN}} = k_{\text{CNN}}\Delta t$, is formed into a matrix in the block Image Former:

$$Y = \begin{pmatrix} y(t - m\Delta t) & \ldots & y(t - \Delta t) & y(t) \\ y(t - \Delta t(m+1)) & \ldots & y(t - 2\Delta t) & y(t - \Delta t) \\ y(t - \Delta t(m+2)) & \ldots & y(t - 3\Delta t) & y(t - 2\Delta t) \\ \ldots & & \ldots & \ldots \\ y(t - \Delta t(k_{\text{CNN}} + m)) & y(t - \Delta t(k_{\text{CNN}} + 1)) & y(t - k_{\text{CNN}}\Delta t) \end{pmatrix}.$$

The thermal portrait formed in the Image Former block is the input for the CNN, which performs the assessment task for the state of CETS at $t = T\_delay$ on the basis of the energy and resource consumption analysis during time $[t - k_{\text{CNN}}\Delta t, t]$. The use of CNN2D is due to the proposed approach to the design of features in $Y$, which allows new patterns to appear, reflecting the influence of the technological process parameters on the criterion (2).

The blocks "Interpretation of Results RNN" and "Interpretation of Results CNN" transfer the normalized values of LSTM and CNN outputs into absolute ones, and also, taking into account the additional data from the environment (coming from the Environmental Data block), form vectors $Y_{\text{RNN}}$ and $Y_{\text{CNN}}$. Their components contain the assessment for CETS state, energy and resource efficiency, and other characteristics depending on the algorithms put into the interpreters. In the "Interpretation

of Results RNN" and "Interpretation of Results CNN" blocks, it is possible to use a fuzzy model to take into account the existing No-factors influencing the CETS description. In the "Neural Network Results Aggregator" block, the outputs "Interpretation of Results RNN" and "Interpretation of Results CNN" are used to conduct generalized analytics for the CETS state—for example, as factors for the fuzzy inference system—and the output of block R goes to the decision-making system with a higher level of the control hierarchy.

NNB learning is conducted separately for LSTM and CNN, but in both cases it requires a sufficient number of examples; thus, it was divided into two stages:

- "coarse adjustment"—prelearning of neural networks using the existing program and mathematical models of GR, MIMCT, and OTF;
- "fine adjustment"—"additional learning" of networks NNB in the CETS operation process.

When the real CETS operates in nominal or close to nominal mode (that is, not in the entire range of parameter variations), a two-stage procedure provides the coarse adjustment of the networks on the mathematical models for the whole range of parameter variations of the technological process. This allows for generating the required number of training datasets for various CETS operating modes and physicochemical, granulometric, lithological, and thermophysical characteristics of ore raw material.

Further "fine adjustment" requires significantly fewer learning examples and is performed continuously during the CETS operation. The application of this approach allows for achieving high accuracy in assessing the state of the technological process and its forecast due to constant additional training of neural networks [31].

The organization of NNB training for "coarse adjustment" is shown in Figure 3, where we indicate:

- M_GR, M_MIMCT, and M_OTF—software modules realizing mathematical models GR, MIMCT, and OTF, respectively;
- $E_{NN\_r}$, $E_M$—CETS energy consumption calculated in the NNB block and based on the mathematical models, $(Y_{RNN} = E_{NN\_r})$;
- LEA$_r$ (learning error analyzer)—analyzer of error $\Delta E_r = \|E_{NN\_r}, -E_M\|$.

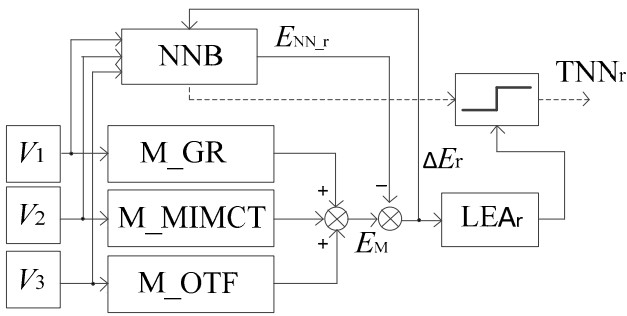

**Figure 3.** Neural network organization with "coarse adjustment".

Vectors of the parameters $V_1$, $V_2$, and $V_3$ are the input of corresponding software modules; they calculate the energy consumption for technological units, after which their total energy consumption $E_M$ is compared with the result $E_{NN\_r}$, given by the NBB block. If mismatching $\Delta E_r$ in the learning process stops decreasing, the LEA$_r$ block opens the threshold element and the trained neural networks (TNN), indicated by the dashed line, go on to the second stage of training, the "fine adjustment" shown in Figure 4.

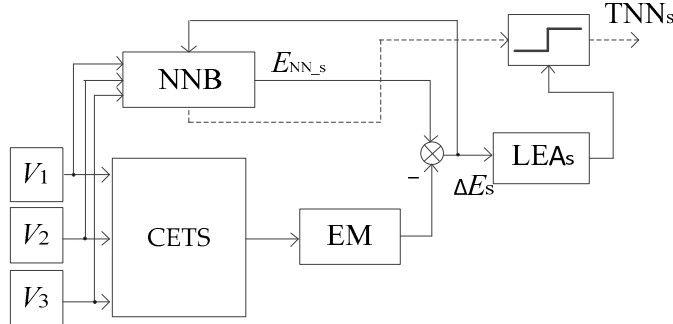

**Figure 4.** Neural network organization with "fine adjustment".

Figure 4 presents: $E_{NN\_s}$, $E_S$—the CETS energy consumption calculated in block NNB ($Y_{RNN} = E_{NN\_s}$) and on the basis of the energy meter (EM) reading, respectively.

LEA$_s$ is the analyzer of the error $E_s = \|E_{NN\_s}, - E_S\|$. If $E_S$ stops decreasing in the learning process, the threshold element is opened up and the neural networks are considered to be ready for use.

After the "fine adjustment," NNB neural networks are ready to optimize energy and resource efficiency for CETS, for which the algorithm is described by the following steps:

1. The formation of optimization parameters set from vectors $V_1$, $V_2$, and $V_3$.
2. Setting the boundaries for the operating ranges of optimization parameter changes and the number of parameter values taken from the ranges.
3. Normalization of the optimization parameter values and formation of a multidimensional coordinate grid, each point of which reflects a certain combination of values for the normalized optimization parameters.
4. Calculation of the value of CETS energy consumption at all points of the multidimensional grid.
5. Selection of a point (or group of points) in a multidimensional coordinate grid in which the minimum energy consumption is achieved; its coordinates are the result of optimization.

In the simplest case, the selection procedure is carried out by a simple enumeration of all energy consumption values; the use of more complex optimization methods is not advisable if there is sufficient computing power.

## 3. Results

Software realization for the presented DTE structure to eliminate monetary costs for the acquisition of development environments is achieved using publicly available resources. Programming language Python 3.6, based on Linux Mint 20 "Ulyana" MATE (64 bits), was chosen by the operating system. When creating and training CNN and LSTM, the open high-level neural network library Keras was used, which is an add-on to the TensorFlow machine learning framework used in this study. The software was run on an ASUS TUF Gaming FX705DT notebook (Version AU039, AsusTek Computer Inc., Taipei, China), AMD Ryzen 7 3750 H CPU, 2.3 GHz, NVIDIA GeForce GTX 1650 4 G GPU, 1024 CUDA cores. Note that the gain time for LSTM training turned out to be significantly less, which is caused by the specific architecture of these networks.

It is not possible to carry out an experiment in DTE due to the lack of a valid CETS sample for the phosphorus production from apatite-nepheline ore waste in Russia. To fill the training sample base, the results of simulation experiments were used on models of individual CETS units [15,32,33]. Their operation results were written into a.csv file, from which the data for learning in NNB were read.

A numerical experiment for testing the proposed DTE structure was carried out with a change in three parameters: the granulometric composition of the input ore raw material $V^1_{TLG,1}$ (as a component of the vector $V^1_{TLG,1}$), the moisture content in a pellet at the output of MIMCT, and the pellet mean diameter. The low dimensionality of variable parameters is used to visualize the results.

The granulometric composition of the input ore raw material (apatite-nepheline ore waste) was characterized by the size of ore particles. A sieve analysis for the tailing dump of JSC Kovdorsky GOK (the Kovdorsky mining and processing plant), according to data from the Mining Institute of the Kola Scientific Center of the Russian Academy of Sciences, showed that, on average, the content of particles of class <0.4 mm is 99%; therefore, in the numerical experiment, the operating range of particle sizes from 0.01 to 0.4 mm was considered. The operating range for moisture content in the pellet $u_0$ was 12–13.5%, the mean diameter for the pellet *d* was in the range from 1.6 to 2.5 cm. In these ranges, 50 points were selected and the harmonic trend of the parameters was set. The total size of the training sample was $10^6$ points, 80% of which were for the training sample; the rest were for the testing one. The structure of the applied neural networks (at $k_{CNN} = 12$, $m = 10$) is shown in Figure 5.

```
Layer (type)           Output Shape        Param #
==================================================
embedding_2 (Embedding)  (None, None, 32)   320000
__________________________________________________
lstm_1 (LSTM)            (None, 32)         8320
__________________________________________________
dense_2 (Dense)          (None, 1)          33
==================================================
Total params: 328,353
Trainable params: 328,353
Non-trainable params: 0
```

```
Layer (type)              Output Shape        Param #
======================================================
conv2d_9 (Conv2D)         (None, 10, 12, 32)   896
______________________________________________________
conv2d_10 (Conv2D)        (None, 10, 12, 32)   9248
______________________________________________________
max_pooling2d_5 (MaxPooling2 (None, 5, 6, 32)   0
______________________________________________________
dropout_7 (Dropout)       (None, 5, 6, 32)     0
______________________________________________________
conv2d_11 (Conv2D)        (None, 5, 6, 64)     18496
______________________________________________________
conv2d_12 (Conv2D)        (None, 3, 4, 64)     36928
______________________________________________________
max_pooling2d_6 (MaxPooling2 (None, 1, 2, 64)   0
______________________________________________________
dropout_8 (Dropout)       (None, 1, 2, 64)     0
______________________________________________________
flatten_3 (Flatten)       (None, 128)          0
______________________________________________________
dense_7 (Dense)           (None, 512)          66048
______________________________________________________
dropout_9 (Dropout)       (None, 512)          0
______________________________________________________
dense_8 (Dense)           (None, 2)            1026
======================================================
Total params: 132,642
Trainable params: 132,642
Non-trainable params: 0
```

(**a**)                                    (**b**)

**Figure 5.** Information about the structure of the applied neural networks: (**a**) Long Short-Term Memory (LSTM) structure; (**b**) convolutional neural network (CNN) structure.

The LSTM network contains 32 internal nodes and solves the regression problem. CNN solves the problem of classifying the status of CETS into two types, operative conditions and faulty conditions. The CNN architecture is defined by the following set of layers and hyperparameters:

- The first cascade of layers:
- The first convolutional layer of Conv2D type works with 2D input data of size $10 \times 12$ (with $k_{CNN} = 12$ and $m = 10$); it has 32 feature maps, and the size of the convolution kernel is $3 \times 3$;
- The second convolutional layer is similar to the first cascade;
- The subsampling layer (MaxPooling2D) with $2 \times 2$ field size;
- The regulation layer, applying the Dropout technique to prevent network relearning, which consists of excluding some of the neurons from the learning process.

  The second cascade of layers is similar to the first cascade;

- The Flatten layer is to reduce the sample size;
- The fully-connected layer of Dense type with 512 neurons and ReLu activation function;
- The Dropout regularizing layer;
- The Dense fully-connected layer with the function of Softmax activation.

Some of the hyperparameters were set to the default, accepted in the Keras framework. Network training was carried out over 100 epochs. The accuracy of the trained networks on the testing sample for LSTM was 95%; for CNN it was 87%.

Figure 6 shows the lines for the level of a two-dimensional cut of the surface for criterion (2) in terms of the parameters $V^1_{TLG,1}$ and $u_0$, and Figure 7 shows the cut level lines according to the parameters $d$ and $u_0$ (at $T\_delay = 0$). The lines form of the level, shown in Figures 6 and 7, indicates the polyextremity of the response surface $E_\Sigma$; therefore, the applied method for simple enumeration of the energy resource efficiency criterion for global optimization is justified in this case.

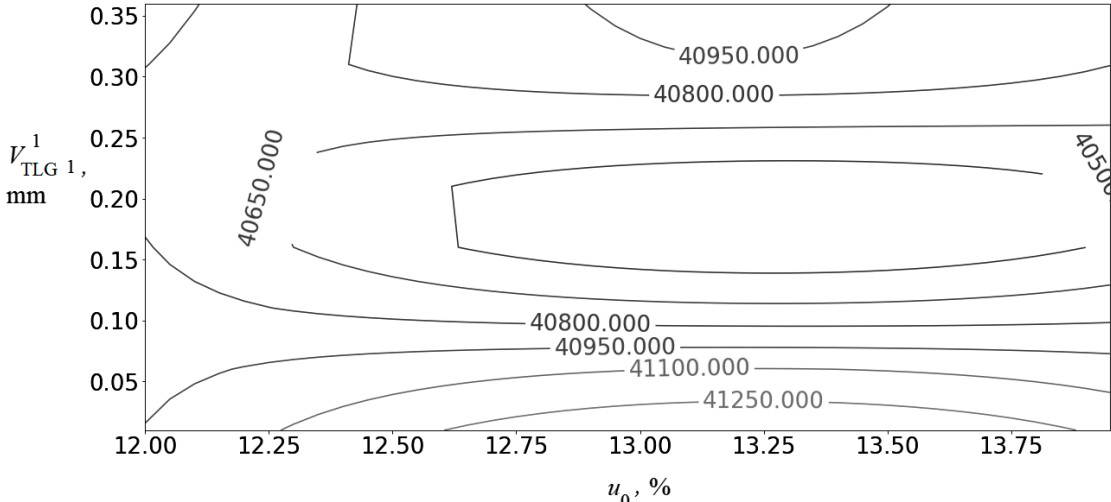

**Figure 6.** $E_\Sigma$ level lines in the cut of parameters $V^1_{TLG,1}$ and $u_0$.

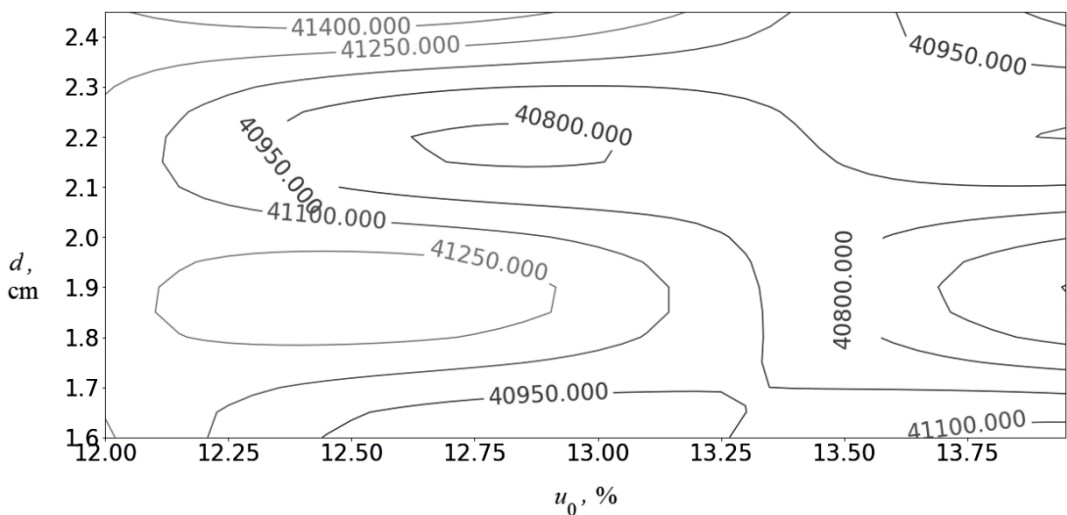

**Figure 7.** $E_\Sigma$ level lines in the cut of parameters $d$ and $u_0$.

The optimal values of parameters are determined during the optimization process. In this example we get the range of values (12.63; 13.11)% for $u_0$; (2.13; 2.24) cm for $d$, and (0.147; 0.239) mm. for $V^1_{TLG,1}$. With an increase in the $T\_delay$ parameter, a decrease in the accuracy of the energy consumption assessment is observed; therefore, when using the forecast results, an additional study should be carried out to find its permissible values.

Figure 8 shows a fragment of the values for the criterion $E_\Sigma$ according to the results of model calculations (indicated by circles) and its forecast (indicated by points) when $T\_delay$ changes from 0 to $5\Delta t$. In Figure 8, the visual analysis shows that the forecasting values $E_\Sigma$ lie close to the exact values, which can indicate the advisability of applying the proposed option to the DTE structure.

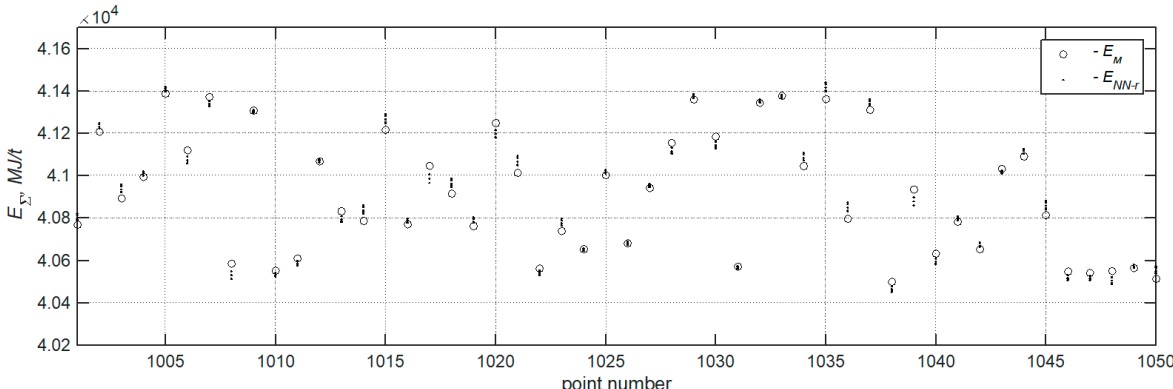

**Figure 8.** Values for the criterion $E_\Sigma$ according to the model calculations (circles) and its forecast in Neural Network Block (NNB) block (points) when parameter *T_delay* is changed from 0 to $5\Delta t$.

Unfortunately, it is not possible to evaluate the quality of the results obtained in comparison with the previous methods, since there is no general optimization model for the entire CETS taking into account the synergetics of the processes. In these conditions, the quality of the results obtained using the presented "digital environment" DTE can be considered a starting point for the analysis of other solutions to optimize the energy and resource efficiency of CETS.

## 4. Conclusions

The structure of the digital environment, presented in this work as an element of the "digital twin" technology, allows for optimizing energy consumption in a complex technological system for the production of phosphorus from apatite-nepheline ore waste. The digital environment is based on such computational intelligence methods as deep neural networks, which make it possible to conduct automated deep analysis for large volumes of technological data. A significant number of optimization parameters leads to a polyextremity of the response surface of the optimality criterion (total energy consumption by the technological system); therefore, to ensure global optimization, simple enumeration of the criterion values at various parameter combinations was used. The contribution of the presented studies to the information support of CETS lies in the developed structure and software of the DTE, which allows for optimizing CETS functioning according to energy and resource efficiency. The results of the numerical experiment demonstrate the capabilities of the created software and the efficiency of the proposed multistage optimization procedure. Further expansion of the DTE functionality is planned to calculate thermodynamic, thermophysical, hydraulic, and other processes in phosphorus production.

**Author Contributions:** Conceptualization, V.M.; methodology, M.D.; software, A.P. and M.D.; validation, I.A.; formal analysis, A.P.; investigation, M.D.; resources, R.A.; data curation, A.P.; writing—original draft preparation, A.P.; writing—review and editing, A.P.; supervision, M.D.; project administration, M.D.; funding acquisition, M.D., I.A., R.S. and R.A. All authors have read and agreed to the published version of the manuscript.

**Funding:** This research was funded by RFBR under the research project No 18-29-24094 and the framework of the state assignment, project number FSWF-2020-0019.

**Conflicts of Interest:** The authors declare no conflict of interest.

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
