# Peer review of "Energy and Resource Efficiency in Apatite-Nepheline Ore Waste Processing Using the Digital Twin Approach"

_energies, doi:10.3390/en13215829_

Round 1

Reviewer 1 Report

Please check the attached file below.

Reviewer 2 Report

The article proposes the structure for DTE CETS software to produce phosphorus from apatite-nepheline ores wastes, which allows optimizing its operating conditions according to the criterion of minimum heat and electrical energy consumption. I have the following comments to improve the article:

 The introduction starts very hastily, I suggest adding more background and also add related research work done in this area. For example, the author can start the introduction by discussing more on Increasing Energy and Resource Efficiency and the impact. The author can also mention recent and relevant work done on increasing Energy and Resource Efficiency. For example:

  1. (2019). Process intensification of solar thermal power using hybridization, flexible heat integration, and real-time optimization. Chemical Engineering and Processing-Process Intensification139, 155-171.
  2. (2019). DURE: An energy-and resource-efficient TCAM architecture for FPGAs with dynamic updates. IEEE Transactions on Very Large Scale Integration (VLSI) Systems27(6), 1298-1307.

Please insert a proper nomenclature to cover all symbols and abbreviations.

I suggest adding equation numbers and also references to all the equations that are not original.

Reviewer 3 Report

his is an interesting paper on an interesting topic in an important area of study.  I do not have any significant concerns about the manuscript.

Author Response

Thanks a lot for the review

Round 2

Reviewer 1 Report

Dear Authors,

Please find the comments for your manuscript in the attached response file.

Thanks for your efforts and contributions.

Author Response

answer in file

Round 3

Reviewer 1 Report

Dear Author, 

  1. Please check the title of your manuscript again. For example, "Energy and Resource Efficiency in Apatite-Nepheline Ores Wastes Processing Using Digital Twin Approach". 
  2. The manuscript must be used extensive English editing in order to be published. 
  3. Check the Figures, Tables order, and alignment again.
  4. Your added sentences in the Abstract: "The main contribution of the ...., is also a contribution of this research into practice." However, We expect we expect to see the most important result in the experiment. Therefore, readers can easy to identify the effectiveness of the proposed method.
  5. We strongly recomend using Table for DNNs models' parameters.

To Editor:

In the second revision round, I have emphasized that this manuscript must be used extensive English editing in order to further proceed.
Therefore, I strongly recommend you remind the author of the English editing service.
